# Conductometric Gas Sensor Based on MoO_3_ Nanoribbon Modified with rGO Nanosheets for Ethylenediamine Detection at Room Temperature

**DOI:** 10.3390/nano13152220

**Published:** 2023-07-31

**Authors:** Hongda Liu, Jiongjiang Liu, Qi Liu, Yinghui Li, Guo Zhang, Chunying He

**Affiliations:** 1Key Laboratory of Functional Inorganic Material Chemistry, School of Chemical Engineering and Material, Heilongjiang University, Ministry of Education, 74 Xuefu Road, Harbin 150080, China; hongdaliuhdl@gmail.com (H.L.); zuibang@163.com (Y.L.); 2School of Chemical Engineering and Material, Heilongjiang University, 74 Xuefu Road, Harbin 150080, China; particularly67@gmail.com (J.L.); laidarao@163.com (Q.L.)

**Keywords:** MoO_3_ nanoribbons, reduced graphene oxide (rGO), EDA, gas sensor, room temperature, selectivity

## Abstract

An ethylenediamine (EDA) gas sensor based on a composite of MoO_3_ nanoribbon and reduced graphene oxide (rGO) was fabricated in this work. MoO_3_ nanoribbon/rGO composites were synthesized using a hydrothermal process. The crystal structure, morphology, and elemental composition of MoO_3_/rGO were analyzed via XRD, FT-IR, Raman, TEM, SEM, XPS, and EPR characterization. The response value of MoO_3_/rGO to 100 ppm ethylenediamine was 843.7 at room temperature, 1.9 times higher than that of MoO_3_ nanoribbons. The MoO_3_/rGO sensor has a low detection limit (LOD) of 0.235 ppm, short response time (8 s), good selectivity, and long-term stability. The improved gas-sensitive performance of MoO_3_/rGO composites is mainly due to the excellent electron transport properties of graphene, the generation of heterojunctions, the higher content of oxygen vacancies, and the large specific surface area in the composites. This study presents a new approach to efficiently and selectively detect ethylenediamine vapor with low power.

## 1. Introduction

Ethylenediamine (C_2_H_8_N_2_, EDA) is an important chemical feedstock widely used in petrochemical and pharmaceutical applications, printing, dyeing, electroplating, and fine chemical intermediates [1,2]. However, EDA is a hazardous chemical that is volatile, corrosive, and flammable, causing environmental pollution and representing a serious threat to human health [3]. EDA vapors can invade the body through the respiratory system and skin, thus causing serious health problems such as conjunctivitis, pneumonia, contact dermatitis, asthma, liver and kidney dysfunction, and even tumors [4,5]. The permissible concentration-short term exposure limit (PC-STEL) for EDA is 10 ppm [6]. Therefore, achieving real-time, effective detection and monitoring of EDA is essential to production life safety. Currently, most methods for detecting EDA rely on expensive and complex instruments such as mass spectrometry, gas chromatography, liquid chromatography, and fluorescence probes [7,8]. Therefore, there is a critical challenge in developing economical, portable, real-time, and selective EDA inspection devices that offer room-temperature operation.

Metal-oxide-based gas sensors (MOS) have attracted increasing attention due to their low cost, small size, high sensitivity, and convenience [9,10]. Orthorhombic MoO_3_ is an n-type semiconductor extensively applied in lithium-ion batteries, catalysts, supercapacitors, photodetectors, and gas sensors due to its unique layer structure, stable physicochemical properties, high dielectric constant, and good catalytic properties [11,12]. In particular, the layered structure of orthorhombic MoO_3_ is suitable for gas sensors due to its easy gas diffusion [13]. Morphology modulation is an effective means of improving the gas-sensitive properties of materials for practical applications [14]. One-dimensional (1D) metal oxides are preferred by researchers due to their high surface activity, large specific surface area, and axial electron transport [15]. Mo et al. [16] synthesized α-MoO_3_ nanobelts via the hydrothermal method and recorded a response value of 174 to 800 ppm ethanol at 300 °C. Bai et al. [15] reported rod-shaped α-MoO_3_ sensing materials, and the response value of the α-MoO_3_ materials was 542 to 500 ppm NO_2_ at 290 °C. Self-assembly α-MoO_3_ nanobelts synthesized by Zhang et al. [17] showed a higher response value to 100 ppm H_2_S, which was 223 at 176 °C. Despite the high response sensitivity of the 1D MoO_3_ sensing materials described above, their high operating temperatures and resulting temperature drift affect measurement accuracy, limiting their application for combustible and explosive gases [18].

Reduced graphene oxide (rGO) with a large specific surface area, extremely high electron mobility, and low electrical noise is considered a promising candidate for gas detection at room temperature [19,20]. Zhang et al. [21] studied the gas sensitivity performance of γ-Fe_2_O_3_/rGO composites with H_2_S at room temperature, and the response value of γ-Fe_2_O_3_/rGO to 97 ppm H_2_S was 520.7. He et al. [22] explored the use of γ-Bi_2_MoO_6_/rGO composites for NO_2_ detection, and the response value of the γ-Bi_2_MoO_6_/rGO composite was 6.4 times higher than that of γ-Bi_2_MoO_6_. The heterojunction between MOS and rGO in the composites can provide a large number of charge carrier transport channels and create more active sites [23]. Moreover, the incorporation of rGO can prevent the agglomeration of MOS nanomaterials and thus increase the total specific surface area [24].

In this paper, MoO_3_/rGO composites were synthesized using a hydrothermal process. The gas-sensitive performance of pure MoO_3_ and MoO_3_/rGO composites toward EDA at room temperature was investigated. Compared to pure MoO_3_ nanoribbons, the gas-sensitive properties of MoO_3_/rGO composites significantly improved. The enhanced gas-sensitive performance mechanism of MoO_3_/rGO composites is also discussed in detail.

## 2. Materials and Methods

### 2.1. Materials

Ammonium heptamolybdate tetrahydrate(AHM, (NH_4_)_6_Mo_7_O_24_.4H_2_O, ≥99.0%) and graphite powder were purchased from Shanghai Maclean Biochemical Co. Nitric acid (HNO_3_, 65–68%), concentrated sulfuric acid (H_2_SO_4_, 95–98%), hydrogen peroxide (H_2_O_2_, 30%), sodium nitrate (NaNO_3_, ≥99.0%), sodium sulfate (Na_2_SO_4_, ≥99.0%), and potassium permanganate (KMnO_4_, 99.5%) were purchased from Liaoning Quanrui Reagent Co. Graphene oxide (GO) was synthesized using a modified Hummers method [25]. 

### 2.2. Preparation of MoO_3_/rGO

In the preparation of MoO_3_/rGO nanocomposites, GO (7 mg) was ultrasonically dispersed in 10 mL of deionized water (DIW) for 2 h to obtain the GO solution. A quantity of 0.618 g of AHM was dissolved in 15 mL of DIW and then added to the above GO solution with continuous stirring for 0.5 h. Subsequently, 2.5 mL of HNO_3_ was added into the mixed solution under stirring conditions and maintained for 0.5 h. The solution was transferred to a 50 mL stainless steel reactor and heated at 180 °C for 20 h. The precipitate was washed several times with DIW and ethanol and then dried overnight at 70 °C in a vacuum oven. Pure MoO_3_ nanoribbons were synthesized without the addition of GO using the procedure described above.

### 2.3. Characterization

The phase structure of the products was determined with an X-ray diffractometer (XRD, Bruker D8A25, CuK, λ = 1.5406 Å). The Fourier-transform infrared spectra (FT-IR) of the samples were obtained via FT-IR spectroscopy (PerkinElmer, Waltham, MA, USA). Raman spectra were recorded using a Raman spectrometer (Horiba, Labram HR800 Evolution, Kyoto, Japan) The morphologies and microstructures of the products were characterized with a cold-field emission scanning electron microscope (FESEM, Hitachi S4800, Hitachi, Tokyo, Japan) and transmission electron microscopy (TEM, Tecnai G^2^ 20, FEI, Billerica, MA, USA). The N_2_ isotherms for adsorption and desorption were determined using the Brunauer–Emmett Teller method (BET, TriStar II 3020, TriStar, Culver City, CA, USA), and the pore size was determined with the Barrett–Joyner–Halenda technique (BJH). The surface chemical element information of the products was investigated via X-ray photoelectron spectroscopy (XPS, Shimadzu Corporation, Kyoto, Japan) with monochromatic radiation of Al Kα (1486.4 eV), and the corrected standard carbon peak was C1s (284.6 eV). The oxygen vacancy content was investigated using electron paramagnetic resonance (EPR, Bruker EMXPLUS, GER, Bruker, Billerica, MA, USA). Experiments with O_2_ temperature-programmed desorption (O_2_-TPD) were performed using an automated chemisorption analyzer (Autochem II 2920, Autochem, Repentigny, QC, Canada).

### 2.4. Sensor Fabrication and Measurement

The as-synthesized MoO_3_ and MoO_3_/rGO powders were ultrasonically dispersed in ethanol, which was applied directly onto Au-interdigitated electrodes (spacing: about 50 μm) through the drop-cast method. After drop casting, the prepared devices were heated at 70 °C for 12 h in a vacuum atmosphere to evaporate the water molecules completely, thus generating a dry thin film to bridge the interdigitated electrodes. 

The static gas-sensing performance of the obtained products was tested using a WS-30B measurement system (China), and the bias voltage of 5 V, relative humidity (RH) of 20 ± 2%, and room temperature (23 ± 2 °C) were controlled during measurements. The operational process and test principles for the sensors are similar to those in previous studies [26]. S = R_a_/R_g_ or S = R_g_/R_a_ represents the response (S) of the sensor, where R_a_ and R_g_ are the resistance value of the sensor in the air and the target gas, respectively. Response/recovery time is defined here as the time required to achieve 90% of the total change in resistance value.

### 2.5. Electrochemical Measurements

All electrochemical tests were performed in a three-electrode system using a CHI660E electrochemical workstation (Shanghai, China) with 0.1 M sodium sulfate (Na_2_SO_4_) as the electrolyte. The saturated Ag/AgCl electrode, Pt electrode, and Glassy carbon electrode (GCE) were used as the reference electrode, counter electrode, and working electrode, respectively. An amount of 5 mg of the sample was dispersed in 2 mL of ethanol and 10 µL of Nafion solution, and then the solution was drop coated on the GCE and dried at room temperature. The prepared electrodes were immersed in the electrolyte for 1 h to ensure that the open-circuit voltage (OCP) was stabilized before starting electrochemical measurements. The electrochemical impedance spectroscopy (EIS) measurement assay settings were as follows: OCP bias 0.34 V, frequency range 0.1–100 kHz, and ac amplitude 5 mV. For the Mott–Schottky (MS) measurements, the increments were 20 mV, the frequency was 1000 Hz, and amplitude was 5 mV.

## 3. Results and Discussion

### 3.1. Structural and Morphological Characterization

The XRD spectra (Figure 1a) of MoO_3_ showed diffraction peaks of 12.8°, 23.2°, 25.8°, 27.4°, 33.7°, 38.0°, 46.3°, 49.2°, and 58.8° corresponding to the orthorhombic phases of MoO_3_ (020), (110), (040), (021), (111), (060), (210), (002), and (081) crystal planes (JCPDS No. 35-0609), respectively. The intensity of the diffraction peaks in the (020), (040), and (060) crystal planes of MoO_3_/rGO were found to be clearly higher than those of MoO_3_, indicating the presence of a layered crystal structure for the anisotropic growth of MoO_3_ in composites [27,28]. The characteristic diffraction peak of rGO was not found in the XRD pattern of MoO_3_/rGO composites. This is mainly attributed to the relatively low content and peak intensity of rGO [29]. Figure 1b indicates the FT-IR spectra of pure MoO_3_ and MoO_3_/rGO composites. Here, the three absorption peaks in the range of 500–1000 cm^−1^ for MoO_3_ at approximately 559, 875, and 998 cm^−1^ correspond to the stretching vibrations of (Mo_3_-O), (Mo_2_-O), and (Mo=O), respectively [11,30]. For MoO_3_/rGO, two new absorption peaks are visible at about 1231 and 1613 cm^−1^ and assigned to the C=C and C-O-C stretching vibrations of rGO [31]. The FT-IR test results confirmed the presence of rGO in the composites. Raman spectroscopy is an effective means of characterizing carbon materials with the usual features of G-band and D-band, and that of MoO_3_ and MoO_3_/rGO is shown In Figure 1c. The characteristic peaks at 994 and 818 cm^−1^ are ascribed to the stretching vibrations of the Mo=O bond, and at 665 cm^−1^, they correspond to the asymmetrical stretching vibrations of the Mo_2_-O bond [32]. The peaks in the range of 100–400 cm^−1^ are related to the various bending modes of α-MoO_3_ crystals [33]. Besides the peaks of MoO_3_, the D (1350 cm^−1^) and G (1598 cm^−1^) characteristic peaks of rGO also exist in MoO_3_/rGO. The D band is assigned to the breathing mode of sp^3^-hybridized carbon, structural defects, and amorphous carbon, whereas the G band corresponds to the scattering mode of sp^2^ carbon [34]. Raman spectroscopy further confirmed the successful preparation of MoO_3_/rGO composites.

SEM images of MoO_3_ and MoO_3_/rGO composites are illustrated in Figure 2a,b. It can be found that the MoO_3_ in both materials consists of nanoribbons. Figure 2c presents the TEM images of MoO_3_/rGO, which clearly show that the folded rGO nanosheets are closely connected to the MoO_3_ nanoribbons. This result further demonstrates the successful fabrication of MoO_3_/rGO composites. Figure 2d provides the HRTEM image of MoO_3_/rGO. Spacing in lattice stripes of 0.198 nm and 0.364 nm was discovered in the HRTEM image, attributed to the (200) and (040) crystal planes of MoO_3_ (JCPDS No. 35-0609), respectively. The selected area electron diffraction (SAED) pattern of the MoO_3_/rGO nanocomposite is displayed in the inset of Figure 2d. Here, the diffraction spots along the orthogonal MoO_3_ [010] zone axes correspond to the diffraction at the (200) and (002) crystal planes [35,36]. Together, the TEM and SAED suggest that orthorhombic MoO_3_ nanoribbons grow mainly along the [001] direction [37].

The full XPS spectra (Figure 3a) of MoO_3_ and MoO_3_/rGO indicate the presence of Mo, O, and C elements in both materials. The C 1s peak at 284.8 eV in the XPS full spectrum of pure MoO_3_ was caused by the C contamination of the analyzer. Figure 3b presents the C 1s high-resolution spectra of MoO_3_/rGO. The fitted peaks at 284.6, 286.1, and 288.7 eV were ascribed to the C-C, C-O, and C=O groups [18,31]. Figure 3c shows high-resolution Mo 3d XPS spectra of pure MoO_3_ and MoO_3_/rGO. The peaks at 233.1 and 236.2 eV are assigned to the binding energies of Mo 3d_3/2_ and Mo 3d_5/2_ orbital electrons of Mo^6+^ [35]. Moreover, the binding energies at about 232.1 eV (Mo 3d_5/2_) and 235.2 eV (Mo 3d_3/2_) correspond to Mo^5+^ [35]. Table 1 presents the relative content of Mo^5+^ and Mo^6+^ in the two materials. The relative content of Mo^5+^ in the composites increased from 4.6% to 9.7% compared to pure MoO_3_. This result demonstrates the higher content of oxygen vacancies in the composites [38]. Figure 3d illustrates the high-resolution spectra of O 1s. The peaks of binding energy around 530.4, 531.4, and 532.5 eV are ascribed to the lattice oxygen (O_L_), oxygen vacancies (O_V_), and chemisorbed oxygen (O_C_) of the samples, respectively [39]. The content of oxygen species in MoO_3_ and MoO_3_/rGO is listed in Table 1. Evidently, the O_C_ and O_V_ content in the composites is higher than that of MoO_3_.

XPS analysis indicated that the complexes contained more oxygen vacancies and chemisorbed oxygen. Both materials were examined via EPR and O_2_-TPD to further examine the content of oxygen vacancies and chemisorbed oxygen. Figure 4a provides the EPR spectra of MoO_3_ and MoO_3_/rGO. Both samples display a Lorentz line (g = 2.001). Here, the signal intensity of the MoO_3_/rGO composite is significantly higher than that of pure MoO_3_. This result means that MoO_3_/rGO has more oxygen vacancies [40]. Figure 4b presents the O_2_-TPD curves of MoO_3_ and MoO_3_/rGO composites. The resolved peaks at lower temperatures (<100 °C) are attributed to physisorbed oxygen, and the peaks at higher temperatures (250–550 °C) are associated with chemisorbed oxygen [41]. Compared to pure MoO_3_, the chemisorbed oxygen peak of the MoO_3_/rGO composite presented a lower resolution temperature and larger peak area. This result illustrates that the chemisorbed oxygen is more active in the composite [42]. The analysis of the EPR and O_2_-TPD spectra is in agreement with the XPS results. It is well known that the presence of more adsorbed oxygen species corresponds to more gas-sensing properties in sensors [43].

The N_2_ adsorption–desorption isotherms of MoO_3_ and MoO_3_/rGO are depicted in Figure 5. The specific surface areas of MoO_3_ and MoO_3_/rGO were 7.7 and 39.3 m^2^/g, respectively. It is evident that MoO_3_/rGO has a larger specific surface area. According to the IUPAC classification, the isotherms of both samples can be classified as type IV isotherms with H3-type hysteresis loops, indicating the presence of mesoporous structures [44]. The average pore sizes of MoO_3_ and MoO_3_/rGO were 3.46 and 3.39 nm, respectively, when calculated by the BJH method. This result demonstrates that MoO_3_ and MoO_3_/rGO are mesoporous materials. In addition, MoO_3_/rGO (0.047 m^3^/g) has a larger pore volume than MoO_3_ (0.037 m^3^/g). The larger specific surface area and rich pore channels of MoO_3_/rGO allow for the exposure of many active sites to interact with the target gas [9,45].

Figure 6a,b show the chemical impedance spectra (EIS) and Mott–Schottky plots (MS) of MoO_3_ and MoO_3_/rGO. The diameter of the semicircle represents the charge transfer resistance (Rct) at the semiconductor–electrolyte interface. The comparison shows that the Rct of the MoO_3_/rGO composites is obviously smaller than that of pure MoO_3_, which suggests that rGO helps to increase the charge migration rate in the composite [46]. The MS test results show that the slope of the curve is positive for both materials. This result illustrates that both MoO_3_ and MoO_3_/rGO have the conductive characteristics of n-type semiconductors.

### 3.2. Gas-Sensing Properties

Figure 7a,d show the response/recovery curves of pure MoO_3_ and MoO_3_/rGO for various EDA concentrations at room temperature, respectively. These curves suggest that the response values of both materials are positively correlated with EDA vapor concentration. The adsorption of reductive EDA on the material surface leads to a decrease in resistance, indicating that both materials have properties typical of n-type semiconductors. Figure 7b,e display histograms of the response values at different EDA concentrations with these two gas sensors. The response values of MoO_3_/rGO were 834.7–1.1 to 100–0.5 ppm EDA, and the response values of MoO_3_ were 435.1–1.03 to 100–1 ppm EDA. Apparently, the MoO_3_/rGO gas sensor exhibited better gas sensitivity at the same concentration. Figure 7c,f depict the response values of these two sensors as a function of EDA vapor concentration. The two fitted equations for response values y and EDA vapor concentration x for the MoO_3_ and MoO_3_/rGO sensors are expressed as (1) and (2), respectively. The fitted correlation coefficients yielded R^2^ values of 0.9966 and 0.9991. These results show that both MoO_3_ and MoO_3_/rGO have a good functional fit with EDA. The response values of the sensors at low EDA concentrations were linearly fitted to calculate the detection limit (inset of Figure 7c,f). The fitted equations for MoO_3_ and MoO_3_/rGO are expressed as (3) and (4), with R^2^ values of 0.9428 and 0.9989, respectively. The LODs of MoO_3_ and MoO_3_/rGO were 0.531 and 0.235 ppm based on the LOD calculation in Equations (5) and (6), respectively [26]. The test results indicate that MoO_3_/rGO has a wider detection range and higher sensitivity than the pure MoO_3_ sensor.
y = 0.0492x^2^ − 0.5241x + 1.0074  R² = 0.9966(1)
y = 0.0821x^2^ + 0.0569x + 1.2799  R^2^ = 0.9991(2)
y = 0.0955x + 0.8805  R² = 0.9428(3)
y = 1.2917x + 0.5283  R² = 0.9989(4)
LOD = 3RMS/K(5)
RMS = (Z^2^/N)^1/2^(6)

K represents the slope of the fitted curve at low concentrations, and Z represents the standard deviation of the response values.

Response/recovery time and repeatability of gas-sensitive materials to target gas are crucial indexes to assess the sensitivity of gas-sensitive sensors. Therefore, the response/recovery curves of the MoO_3_ and MoO_3_/rGO sensors were compared for 100 ppm EDA gas at room temperature (see Figure 8a,b). Response/recovery time for the MoO_3_ and MoO_3_/rGO sensors to 100 ppm EDA gas were 18/901 s and 8/357 s, respectively. The MoO_3_/rGO sensor yielded a shorter response/recovery time than the pure MoO_3_ sensor. Figure 8c,d show the cyclic response curves for MoO_3_ and MoO_3_/rGO to 100 ppm EDA gas at room temperature. The sensor response values do not vary significantly over the five cycles, indicating that both sensors have good cycling stability.

Selectivity and long-term stability are key parameters for measuring sensor performance in practical applications. The results of the selectivity tests for the MoO_3_ and MoO_3_/rGO gas sensors are plotted in Figure 9a. Obviously, the gas response values of MoO_3_ and MoO_3_/rGO sensors are much higher for 100 ppm EDA gas than for the other interfering gases (100 ppm triethylamine, NH_3_, ethanol, formaldehyde, and acetone gas). This result demonstrates the excellent selectivity of MoO_3_ and MoO_3_/rGO sensors. The long-term stability of MoO_3_ and MoO_3_/rGO tests was evaluated once a week for eight weeks, and the results are summarized in Figure 9b. The response values of MoO_3_ and MoO_3_/rGO varied less than 5% over time, indicating that the sensors have excellent long-term stability. The response and recovery times of these two sensors during long-term stability tests are shown in the inset of Figure 9b. The response and recovery time of the MoO_3_ sensor in general increased with the increase in the test period, which is caused by the agglomeration of the MoO_3_ nanoribbons. The response and recovery times of the MoO_3_/rGO sensor fluctuated in magnitude, though not significantly, compared to the first week. MoO_3_/rGO is more stable compared to MoO_3_ nanoribbons, which is mainly due to the incorporation of rGO, which can effectively prevent the agglomeration of nanoribbons in the composites. Overall, the MoO_3_/rGO sensor showed high response values, good function matching, fast responses, excellent selectivity, and long-term stability for EDA gas detection with potential for practical applications.

We compared the MoO_3_/rGO sensor with other sensors used for EDA detection (Table 2). The MoO_3_/rGO sensor has good gas-sensitive performance for EDA with high response values and low detection limits. The MoO_3_/rGO sensor fabricated in this work is potentially valuable for industrial applications.

### 3.3. Gas-Sensing Mechanism

The resistance changes in the gas-sensitive characteristics of metal oxide semiconductors arise from the chemisorption and desorption of gases on the surfaces of materials [22]. The MoO_3_/rGO composite exhibited an n-type nature in performance tests measuring sensitivity to EDA vapor. Therefore, the resistance changes in MoO_3_/rGO are caused by variation in the electron concentration in the material [54]. Figure 10 illustrates the sensing mode of MoO_3_/rGO composites in the air and the EDA vapor. When the MoO_3_/rGO sensor was exposed to the air atmosphere, O_2_ molecules in the air were adsorbed onto the composite surface and captured electrons from the material to form adsorbed oxygen species ($O_{2}^{-}$) (Equations (7) and (8)) [15]. Simultaneously, an electron depletion layer formed on the MoO_3_ surface, causing a decrease in the charge carrier density of the composite and increasing sensor resistance [29]. When the sensor was exposed to reduced EDA vapor, the EDA molecules adsorbed onto the material’s surface and interacted with the adsorbed oxygen species (Equation (9)) [55]. Meanwhile, the electrons were released from the reaction and back into the material, thereby inducing a decrease in the thickness of the depletion layer and reducing the resistance of the composites [43]. When the sensor returned to the air, O_2_ molecules were adsorbed back onto the surface of the nanobelts. This caused the electron depletion layer to rebuild and the resistance to return to its initial value.
O_2(gas)_ → O_2(ads)_(7)
(8)$$O_{2(\mathrm{ads})}+e^{-}\to O_{2(\mathrm{ads})}^{-}$$
(9)$$C_{2}H_{8}N_{2}+4O_{2(\mathrm{ads})}^{-}\to 2CO_{2}+4H_{2}O+N_{2}+4e^{-}$$

The excellent gas-sensitive performance of MoO_3_/rGO with EDA vapor at room temperature can be attributed to three main factors. First, 1D MoO_3_ features a layered structure formed by the alternating stacking of octahedral MoO_6_ bilayer planes in the [010] direction, and the [010] crystal planes of MoO_3_ nanoribbons have higher catalytic activity [27]. This facilitates the adsorption and diffusion of gas molecules, exposes more active sites, and provides a fast transport path for electrons along the axial direction, thus improving gas-sensitive performance [56]. Secondly, rGO nanosheets can prevent the stacking and agglomeration of MoO_3_ nanobelts, and rGO itself has high electron mobility, which gives the composites a larger specific surface area, more abundant pore channels, and higher electron transport capacity [24]. This not only provides more adsorption sites and effective diffusion pathways for EDA gases but also shortens the response/recovery time, further improving the sensing performance [20]. Third, heterogeneous structures are formed between the two materials when MoO_3_ is combined with rGO. The work function of the n-type material MoO_3_ (5.3 eV) [57] is different from that of the p-type material rGO (4.8 eV) [19]. The electrons in rGO are transferred to MoO_3_ to balance the Fermi energy level (Ef), which causes the energy band to bend and increases the electron concentration of MoO_3_ in the composite [58]. This process allows for more electrons to be trapped by O_2_ molecules adsorbed on the MoO_3_ surface, thereby forming more adsorbed oxygen species and providing more active sites for the material [21,59]. Hence, the gas-sensitive performance is considerably enhanced.

## 4. Conclusions

In summary, MoO_3_/rGO composites were fabricated using a hydrothermal method to develop the first MOS-based resistive gas sensor for the detection of EDA gases. At room temperature, the MoO_3_/rGO composites exhibited higher response values (834.7), shorter response/recovery times (8/357 s), and lower detection limits (0.235 ppm) for EDA compared to pure MoO_3_ nanobelts. In addition, the MoO_3_/rGO composites exhibited good selectivity and long-term stability. The outstanding gas-sensitive performance of this sensor mainly contributed to the formation of heterojunctions between MoO_3_ nanoribbons and rGO alongside the large specific surface area, abundant oxygen vacancies, and good electron transport properties of rGO. This study provides a new direction for the design and application of highly selective and responsive ethylenediamine sensors at room temperature.

## Figures and Tables

**Figure 1 nanomaterials-13-02220-f001:**
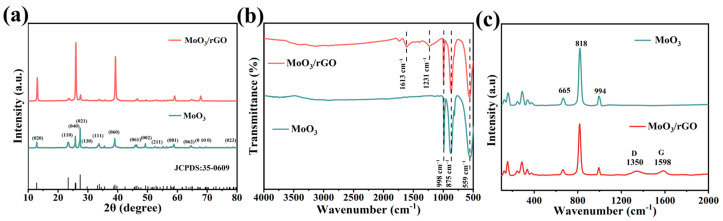
(**a**) XRD patterns and (**b**) FT-IR spectra. (**c**) Raman spectra of MoO_3_ and MoO_3_/rGO.

**Figure 2 nanomaterials-13-02220-f002:**
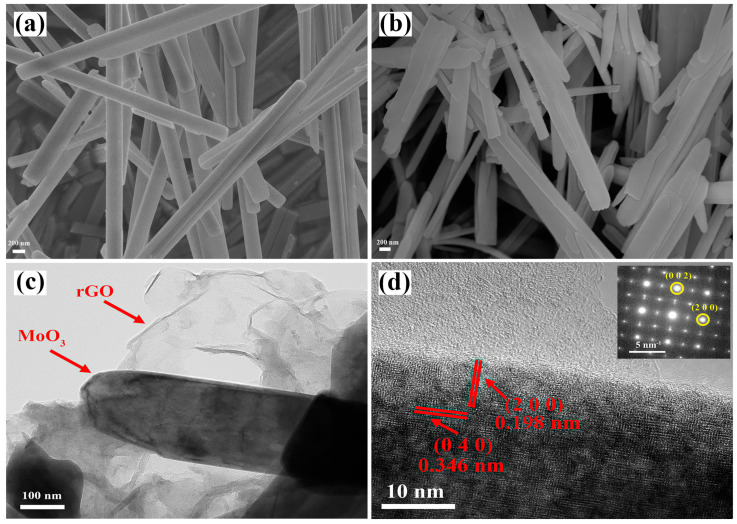
SEM images of (**a**) pure (**b**) MoO_3_/rGO. TEM (**c**) and HRTEM (**d**) images of MoO_3_/rGO (SAED pattern inset (**d**)).

**Figure 3 nanomaterials-13-02220-f003:**
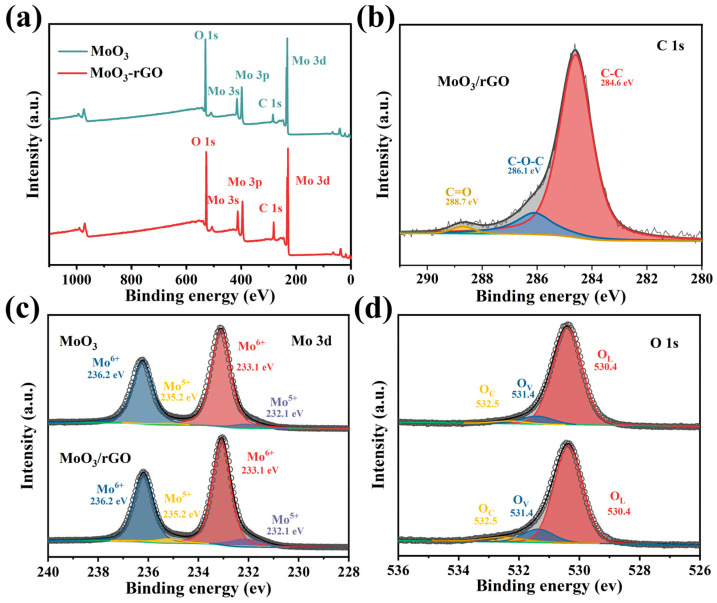
(**a**) XPS spectra of MoO_3_ and MoO_3_/rGO. (**b**) C 1s spectra of MoO_3_/rGO. XPS spectra of (**c**) Mo 3d and (**d**) O 1s for MoO_3_ and MoO_3_/rGO.

**Figure 4 nanomaterials-13-02220-f004:**
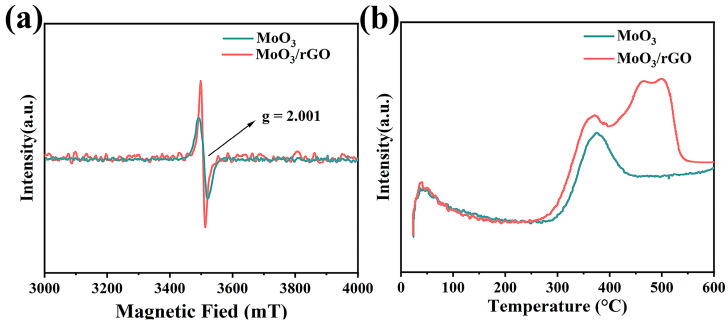
EPR spectra (**a**) and O_2_-TPD spectra (**b**) of MoO_3_ and MoO_3_/rGO.

**Figure 5 nanomaterials-13-02220-f005:**
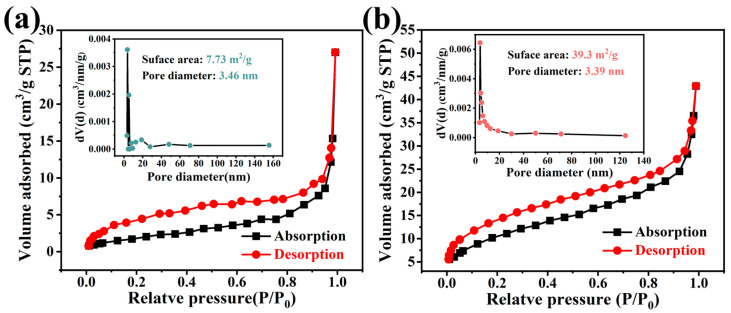
N_2_ adsorption–desorption isotherms (inset: pore size distribution) of (**a**) MoO_3_ and (**b**) MoO_3_/rGO.

**Figure 6 nanomaterials-13-02220-f006:**
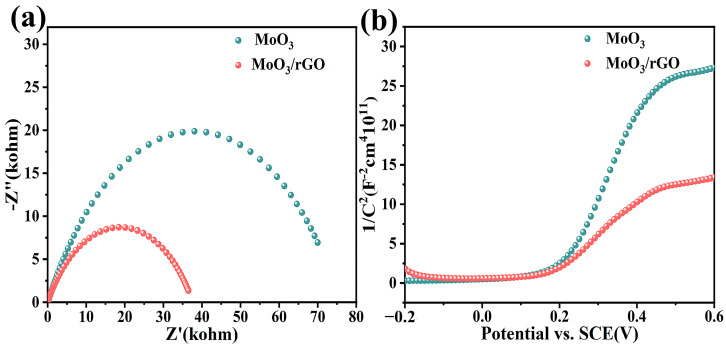
(**a**) Nyquist EIS plots and (**b**) Mott–Schottky plots of MoO_3_ and MoO_3_/rGO.

**Figure 7 nanomaterials-13-02220-f007:**
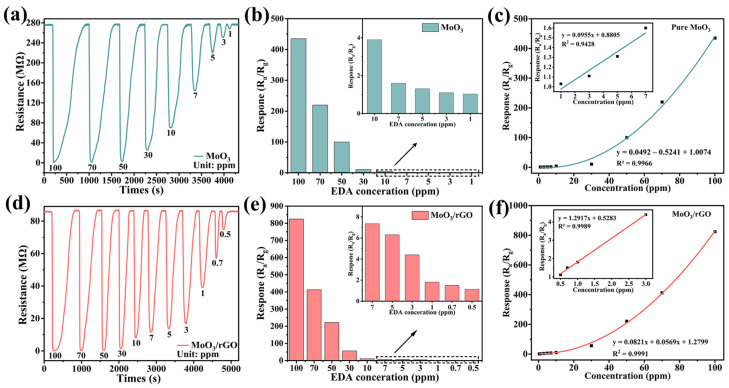
Resistance change curves of (**a**) MoO_3_ and (**d**) MoO_3_/rGO sensors to various EDA concentrations. Response value histograms of (**b**) MoO_3_ and (**e**) MoO_3_/rGO sensors in various EDA concentrations. The relationship between the response values of (**c**) MoO_3_ and (**f**) MoO_3_/rGO sensors with EDA concentrations.

**Figure 8 nanomaterials-13-02220-f008:**
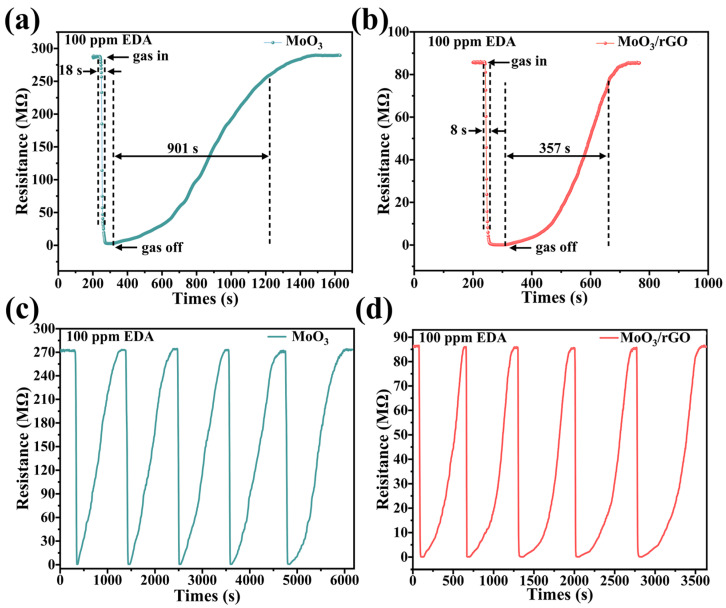
Response/recovery time of (**a**) MoO_3_ and (**b**) MoO_3_/rGO gas sensors. Cyclic response curves of (**c**) MoO_3_ and (**d**) MoO_3_/rGO sensors.

**Figure 9 nanomaterials-13-02220-f009:**
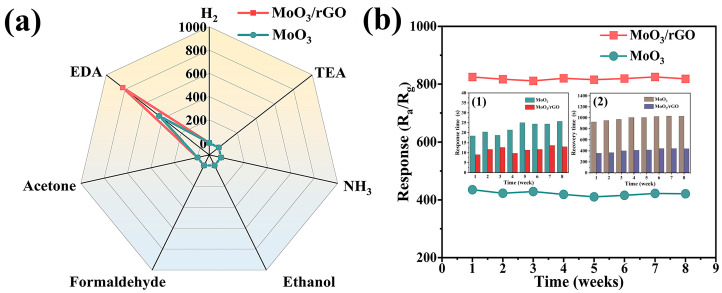
(**a**) Selectivity of MoO_3_ and MoO_3_/rGO sensors to 100 ppm various gases. (**b**) Long-term stability of MoO_3_ and MoO_3_/rGO sensors to 100 ppm EDA gas (response time (1) and recovery time (2) of these two sensors during long-term stability tests inset (**b**)).

**Figure 10 nanomaterials-13-02220-f010:**
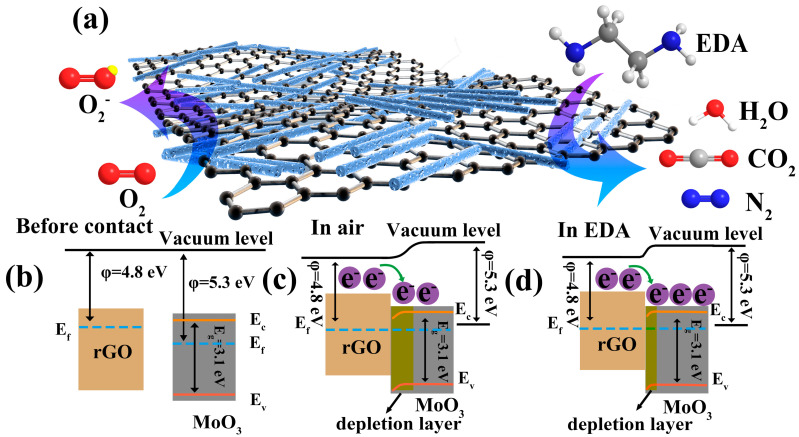
(**a**) Schematic diagrams of the gas-sensing mechanism for MoO_3_/rGO sensors for EDA. (**b**) Band diagram of the rGO and MoO_3_ before contacting. Band diagram of the MoO_3_/rGO composites (**c**) in air and (**d**) in EDA gas.

**Table 1 nanomaterials-13-02220-t001:** The relative content of Mo^5+^, Mo^6+^, and oxygen species in MoO_3_ and MoO_3_/rGO.

Samples	Mo^5+^ (%)	Mo^6+^ (%)	O_L_ (%)	O_V_ (%)	O_C_ (%)
MoO_3_	4.6	65.4	90.9	6.2	2.9
MoO_3_/rGO	9.7	90.3	80.3	11.9	7.8

**Table 2 nanomaterials-13-02220-t002:** Comparison of the EDA-sensing properties of the MoO_3_/rGO in this work and previous reports.

Materials	Method	EDA (ppm)	Response	LOD (ppm)	Ref.
Nitrated polythiophen	Colorimetry	5010	1.99 (A/A_0_)	5.6	[47]
Perylene bisimide	Fluorescence	85.2	1.39 (I_0_/I)	4.0	[48]
[Zn_4_(HIDCPy)_4_(DMSO)(DMF)_3_]_n_	Fluorescence	450	1.77 (I/I_0_ − 1)	3.9	[49]
MP@MOP	Optical	80	0.17 (∆Abs)	15	[50]
OPTA-MSA	Fluorescence	50	80.3 (∆E)	0.70	[5]
Zn_2_(bcpBTD)_2_(bpBTD)(H_2_O)_2_]·DMF(1)	Fluorescence	59.8	3.8 (I/I_0_)	0.052	[3]
[Cd(H_2_L)_2_]·3H_2_O·2DMF	Resistance	900	45%	/	[51]
{[Cd(L)(glu)]·3H_2_O}∞	Fluorescence	2500	3.7 (I_0_/I)	64.5	[52]
Eu@IsoMe@Cu/Ni fabric	Optical	200	51.1	4.74	[53]
MoO_3_/rGO	Resistance	100	834.7	0.235	this work

## Data Availability

Data will be made available on request.

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
