# Peer review of "Conductometric Gas Sensor Based on MoO3 Nanoribbon Modified with rGO Nanosheets for Ethylenediamine Detection at Room Temperature"

_nanomaterials, 2023, doi:10.3390/nano13152220_

Round 1
Reviewer 1 Report
The authors prepared a chemiresistive sensor for ethylenediamine (EDA) detection with a MoO3/rGO composite, evaluated their sensing performances in the points of responsivity, selectivity, and durability, and discussed the sensing mechanism. I think the manuscript can be published on nanomaterials after revisions regarding the following points.
1. In the experimental section, please add the information on the X-ray source (e.g. Al Ka or Mo Ka) for XPS measurements and the binding energy correction.
2. Please add the information on the bias voltage applied to the sensor during the sensing measurements.
3. Measure a gas sensing property of rGO as a control experiment.
4. The resistance values in the EDA concentration of over 50 ppm seem to saturate. Do the sensors significantly distinguish these high concentrations of EDA? Could you add error bars in Figure 10 to show that a significant difference in sensor responses can be observed?
5. Could you add discussions and comparisons about the response and recovery times of the sensors after one to eight weeks from the preparation?
6. The gas sensing mechanism shown in section 3.3 differs from the one discussed in reference 53. According to Ref. 53, O2−(ads) is further reduced to 2O−, then O− reacts with EDA as a following reaction scheme. EDA + 8O− -> 2CO2 + 4H2O + N2 + 8e−. Please add a discussion of why the authors employed the reaction scheme shown in the main text. In addition, the coefficient of N2 should be corrected.
7. The term “p-type material rGO” in line 322 will confuse readers because the band diagram of rGO drawn in Figure 10 looks like a metal.
8. The chemical structure of EDA shown in Figure 10 is not ethylenediamine but 1,1-diaminoethene. And, please add captions for (a) to (d).
Reviewer 2 Report
The author describes the “Conductometric gas sensor based on MoO3 nanoribbon modified with rGO nanosheets for ethylenediamine detection at room temperature”. This original article is quite interesting from a technological point of view. The author should revise their manuscript based on the comments and suggestions. I recommended a Major revision of the manuscript.
The Major suggestion below:
- In the abstract, the author can state a clear research question to convey the main objective of this study.
- The figure quality is too poor and the author should improve the quality of the figure.
- Why author choose rGO for forming a composite? Compare to other carbon-related materials like AC and graphene.
- The TEM, SAED pattern, and XRD must be supporting each other, especially in the prospect of planes with d-spacing.
- The author should provide the Raman spectra of both samples for confirming the existence of rGO in the composite system.
- The author should re-check the XPS spectra of the samples.
- The author should mention the IUPAC classification and hysteresis loop type in the BET analysis.
- The author should provide the EIS measurement details in the revised manuscript.
- The author should improve the grammatical and typo errors in the paper.
Minor editing of English language required.
Reviewer 3 Report
The article “Conductometric gas sensor based on MoO3 nanoribbon modified with rGO nanosheets for ethylenediamine detection at room temperature” by Hongda Liu et al is devoted to synthesis and study of new gas sensors based on MoO3 nanoribbon/reduced graphene oxide composites.
The authors used a set of methods (XRD, FT-IR, TEM, SEM, XPS…) to characterize the crystal structure, morphology, and elemental composition of the MoO3/rGO composite. The MoO3/rGO sensor has been shown to have a low detection limit of 0.235 ppm, short response time, good selectivity and long-term stability.
The article should improve the level of English (in some places articles are missing, there are repetitions).
In general, the article is a good comprehensive study and is recommended for publication in its present form.
The article should improve the level of English (in some places articles are missing, there are repetitions).
